# Endophytic Bacteria with Potential Antimicrobial Activity Isolated from *Theobroma cacao* in Brazilian Amazon

**DOI:** 10.3390/microorganisms13071686

**Published:** 2025-07-18

**Authors:** Lívia Freitas da Silva Pinto, Taynara Cristina Santos Tavares, Oscar Victor Cardenas-Alegria, Elaine Maria Silva Guedes Lobato, Cristina Paiva de Sousa, Adriana Ribeiro Carneiro Nunes

**Affiliations:** 1Laboratory of Microbiology and Biomolecules, Department of Morphology and Pathology, Federal University of São Carlos, São Carlos 13565-905, Brazil; prokarya@ufscar.br; 2Center for Valorization of Amazonian Bioactive Compounds, Federal University of Pará, Belém 66075-110, Brazil; naratavares51@gmail.com (T.C.S.T.); carneiroar@gmail.com (A.R.C.N.); 3Laboratory of Bioinformatics and Genetics of Microorganisms, Institute of Biological Sciences, Federal University of Pará, Belém 66075-110, Brazil; bioscar@gmail.com; 4Laboratory of soil Science, Federal Rural University of Amazon, Paragominas 68627-451, Brazil; elaineguedes1@gmail.com

**Keywords:** actinobacteria, antimicrobial activity, endophytic bacteria, *Theobroma cacao*

## Abstract

Endophytic bacteria inhabit plant tissues without damaging them and have specialized adaptation capabilities that allow them to establish themselves in this ecological niche. Endophytes produce numerous secondary metabolites with antimicrobial, anticancer, and pesticide properties, among others. In this study, endophytic bacteria were isolated and characterized from cocoa plants in a Brazilian municipality, with the view to evaluate their potential antagonistic activity on clinical bacterial strains. The isolates were identified through phenotypic analysis and molecular characterization. After bacterial isolation, it was possible to verify the presence of 11 different endophytic strains, with a bacterial load of up to 6.3 × 10^3^ CFU/g in each plant. The morphological and biochemical profile of the isolates varied. At the taxonomic level, these bacteria showed 99% similarity with the genera Microbacterium, Curtobacterium, Pseudomonas, Bacillus, Ralstonia, and Methylobacterium. The strains of the phylum Actinobacteria, which are known for producing natural bioactive compounds with high biotechnological potential, were effective in inhibiting Staphylococcus aureus ATCC and multidrug-resistant clinical strains. This work aims to expand knowledge about endophytes, with the aim of applying them in other sectors, such as the production of compounds against resistant human pathogens.

## 1. Introduction

Endophytic bacteria are microorganisms, which spend all or part of their lives in plant tissues without causing any apparent damage to them [1]. Nearly all plants are thought to associate with one or more endophytic microbes [2]. However, only a few of these species have been studied for the presence of such microorganisms [3].

Endophytes originate from the existence of an environmental microbiome, coming from the rhizosphere or phyllosphere [4]. Although they may enter plants by the leaves, via stomata, the vast majority begin the colonization process through the roots [5]. Plant roots exude specific compounds (e.g., organic acids, amino acids, and flavonoids), which are recognized by rhizospheric microorganisms and trigger the migratory movement towards the interior of the plant [6].

The plant–endophyte interaction is characterized as symbiotic, as both benefit from this association [7]. The host plants house and protect the endophytes, which in turn promote the growth of plants via nitrogen fixation, phosphorus enrichment, and the synthesis of phytohormones [8]. Furthermore, they act in plant protection as biocontrollers [9], due to competition for nutrients and space; synthesis of lytic enzymes; production of other metabolites with antibiosis activity [10]; tolerance to salinity, temperatures, heavy metals, and contaminated chemicals; and other abiotic factors [11].

The development of these activities results in a specialized adaptation capacity that allows for the establishment of endophytes in this ecological niche [12]. Genomic studies related to the adaptation of bacteria to the endosphere are being developed, and it has been observed that, in general, they present a decrease in the genome [11]. In some cases, endophytes also have large plasmids, with genes involved in the synthesis and transport of amino acids that are expressed in response to changes in environmental conditions and the occurrence of stresses [13].

Thus, the metabolic profiling of endophytes in recent years has identified a wealth of secondary metabolites, including compounds such as flavonoids, carotenoids, melatonins, terpenoids, phenolics, alkaloids, and peptides [14]. These natural molecules offer prospects for a wide range of applications (antimicrobial, anticancer, antioxidant, cytotoxic, pesticide properties) and are the basis for the synthesis of effective molecules in the medical, pharmaceutical, agricultural, and other industrial sectors [15].

On the other hand, actinobacteria are considered a promising source of natural and structurally unique antimicrobials, many of which have potential for therapeutic novelties [15,16], also reported in medicinal plant endophytes [17]. Currently, actinobacteria account for approximately 80% of the synthesis of antibiotics of microbial origin, making a significant contribution to the pharmaceutical industry [18].

The search for and discovery of natural bioactive molecules with novel therapeutic mechanisms is decreasing compared to the increase in emerging infectious diseases and multidrug-resistant human pathogens [19]. However, the prospects for discovering new drug molecules can be improved by moving research to underexplored or unexplored environments [15].

Even so, most studies carried out on plants of commercial interest, such as Theobroma cacao, show that isolated endophytes are pathogenic and may put cocoa agricultural production at risk [20,21,22]. Furthermore, in the Brazilian Amazon, there is little research on the association of endophytes with this plant compared to the great plant heterogeneity existing in the region [5].

In this sense, the purpose of this work was to isolate and characterize endophytic bacteria from leaves and stem fragments of cocoa trees in order to evaluate the potential production of compounds with antagonistic activity in bacterial strains of clinical interest.

## 2. Materials and Methods

### 2.1. Collection Site

The samples were collected from a *Theobroma cacao* plantation (2°59′37″ S and 47°44′44″ W) located in the municipality of Paragominas. This site belongs to the southeastern mesoregion of Pará, Northern Brazil. Cocoa cultivation on this property is divided into two distinct areas: shaded and full-sun cocoa planting. Three vegetables from each planting were randomly chosen for analysis, and 12.5 g of leaves and 12.5 g of stem fragments were collected [23]. The collected material was labeled, stored in thermal boxes at 4 °C, and transported to the laboratory.

### 2.2. Isolation of Bacteria

A mixture of leaves and stem fragments from each plant collected was initially made to isolate the bacteria. Then, each of these underwent a superficial disinfection pre-treatment, based on washing with running water; sequential immersion in 70% ethyl alcohol for 1 min, 2% sodium hypochlorite for 4 min, and 70% ethyl alcohol for 30 s; and washing in sterile distilled water three times. Subsequently, 25 g of tissue fragments was macerated in 225 mL saline solution (NaCl 0.9%). The macerate was stirred at 200 rpm for 15 min and left to rest for another 15 min; this process occurred at room temperature. After that, the mixture was filtered using sterile gauze. From the liquid obtained from the filtrates, serial dilutions of 10^−2^, 10^−4^, and 10^−6^ were carried out in triplicate of them [24].

The bacteria were isolated using the streak depletion method in two culture media supplemented with Nystatin (1 μL/mL): Yeast Extract–Malt Extract agar (ISP-2) and Brain Heart Infusion (BHI) broth. They were then incubated in Petri dishes in a bacteriological oven for 25 days at 28 °C [23].

After incubation, the microorganisms were characterized macroscopically and microscopically by performing Gram staining. The biochemical profile of the isolates was analyzed by testing the production of catalase, growth from citrate as the primary source of energy, the ability to form phenyl pyruvic acid through phenylalanine, the fermentation of carbohydrates (sucrose, glucose, and lactose), and the production of hydrogen sulfide and carbon dioxide [25].

### 2.3. Molecular Identification

Molecular characterization was performed by extracting total genomic DNA from bacteria using the Phenol/Chloroform/Isoamyl Alcohol method [26]. Subsequently, the concentration and purity index of the genetic material were determined using a NanoDrop 2000c device (Thermo Fisher Scientific, Wilmington, DE, USA). The integrity of the genetic material was verified by electrophoresis on 1% agarose gel in 1X TAE buffer (40 nM Tris-acetate; 1 mM EDTA) and stained with ethidium bromide at 0.5 μg/mL. After that, the 16S rRNA region was amplified with the help of universal primers 8F and 1492R [27]. 

The products obtained by amplification were sequenced using the Sanger method. Sequence visualization was performed using the BioEdit 7.0.5 program [28]. With the same tool, multiple alignment was performed with the ClustalW method, comparing the sequences obtained with those from the public GenBank database (http://www.ncbi.nlm.nih.gov, accessed on 13 January 2024). The J Model Test 2.1.10 software [29] was initially used to develop the phylogenetic tree to select the best statistical method and model for building the tree with the help of the MEGA 11.0.10 program [30].

### 2.4. Antimicrobial Evaluation

The growth curve was carried out on the isolates belonging to the actinobacteria phylum to identify the stationary phase, where there is greater production of secondary metabolites [31]. The potential antimicrobial activity of the bacteria was evaluated using the diffusion test in solid media, carried out in triplicate.

The testing occurred by comparing the isolates with nine strains, 2 of which correspond to the reference strains *Staphylococcus aureus* (ATCC-25923) and *Escherichia coli* (ATCC-25922), and 7 to clinical strains of multidrug-resistant *Staphylococcus aureus*, *Escherichia coli*, *Klebsiella pneumoniae*, *Pseudomonas aeruginosa*, *Enterobacter cloacae*, *Acinetobacter baumanii*, and *Serratia marcescens*. The clinical strains were obtained from biological samples from João de Barros Barreto University Hospital, after approval by the Human Research Ethics Committee with procedural number CAAE78135723.6.0000.0017.

This methodology added a bacterial inoculum with a turbidity of 0.5 on the McFarland scale of ATCC and clinical strains in the exponential growth phase to Mueller–Hinton agar. After sowing, each strain using the distension method on a plate, 6 mm diameter wells were created on the surface of the seeded medium, and ISP-2 solid medium was added to each well with actinobacteria growing in the stationary phase. These plates were placed in an oven at 37 °C for 48 h to visualize bacterial development later and verify the inhibition of halo formation [32].

## 3. Results and Discussion

### 3.1. Characteristics of Isolated Strains

Of the six samples of leaves and stem fragments collected, 19 endophytic bacterial colonies were obtained, 14 (73.68%) referring to the three cocoa trees grown in full sun and 5 (26.32%) isolated from plants in a shaded area. Regarding the number of colonies per sample, values between one and seven were observed, except for one sample that showed no growth. In this way, the bacterial load obtained was up to 6.3 × 10^3^ CFU/g, with significant differences between vegetables with and without shade (*t*-test = 4.943; *p* value = 0.004), with the load being more important in planting in full sun.

However, the sample size used in this study was insufficient to accurately determine the quantitative superiority of endophytes planted in full sun. In view of this, future studies with a larger number of samples are needed to compare these two types of cultivation.

Numerous factors can influence the colonization and promote population fluctuations in endophytes, including environmental conditions such as sunlight, water activity, temperature, and humidity [33]. Andrade et al. [34] evaluated the diversity of endophytic bacteria from *Cattleya walkerina* in three different environments (nature, greenhouse, and in vitro), finding differences between them, which highlights the influence of environment on the colonization capacity of endophytic bacteria.

On the other hand, the number of endophytic isolates was lower than that reported in the research by Alsultan et al. [35], which isolated up to 12 endophytic bacteria in each leaf sample. This difference in our work may be due to the mixing of stem fragments, which did not allow for homogeneous maceration, as did the work of Vera-Loor et al. [36], carried out with flower and cocoa seed tissues.

In addition, the number of endophytes of the same plant can be variable in different tissues; values found in root samples were significantly higher than those in stems and leaves [37]. On the other hand, antifungal activity tested against *Ganoderma boninense* was more relevant in isolates from higher tissues of the plant [38].

The characteristics of isolates were variable and can be seen in Table 1. The colonies presented different sizes and shapes, with colors that varied from white to pink, yellow, and non-pigmented, some with shine and others without. Smooth and rough textures with a soft or mucoid consistency were also found (Figure 1).

In the results of our study, different colors were observed in the colonies, which may suggest the presence of pigments, possibly carotenoids, as these molecules were identified in yellowish colonies [39,40,41,42]. Similar characteristics of bacterial colonies were also found in isolates of *Citrullus colocynthis* (L.) Schrad with ISP-2 medium [43].

Furthermore, of the 19 colonies found, 15 isolates with different macroscopic characteristics were identified. However, only 11 were subjected to Gram staining, as 4 strains were lost in the process of replication and purification of the colonies.

Regarding Gram staining, the presence of bacilli was evident in all isolates. The variation in micromorphology was restricted to dye affinity, as five of the total isolates were Gram-positive. The majority of endophytic bacteria are Gram-positive, but it is also possible to find Gram-negative endophytic bacteria [44].

In plants of *Ficus minhassae*, the isolation was similar in Gram-positive and Gram-negative endophytic bacteria, classified as *Actinobacteria* and *Proteobacter*, respectively [44]. In contrast, in another study more Gram-negative strains were found [45].

The biochemical characteristics of the Gram-positive strains of catalase-positive, phenylalanine-negative, and non-H_2_S producing samples were similar in almost all isolates except isolate CP6A01, which showed different results in the use of citrate as a carbon source and fermentation of glucose, sucrose, and lactose. Also, variation was observed in the other isolates regarding CO_2_ production.

Important Gram-positive bacteria that are beneficial for plant growth showed positive catalase and citrate results, resembling the strain CP6A01 [46]. Moreover, the variation in sucrose and lactose fermentation observed in this same strain was reported in another study [47].

In the case of Gram-negatives, it can be seen that the biochemical profile was phenylalanine positive and non-H_2_S producing, with a difference in glucose fermentation being observed. Furthermore, strain CP1A04 showed different results from the others regarding catalase activity. In the same way, most Gram-negative endophytic bacteria can produce catalase, while a small proportion are considered catalase-negative [48].

The distinct biochemical profiles found in bacterial isolates may be atypical, as they depend on their genetic load, which is associated with the ecological niche they inhabit [49].

### 3.2. 16S rRNA Analysis

Identification with the sequencing of the 16S rRNA segment showed 99% similarity with reference sequences from the database, obtaining nominations at the taxonomic level of genus; these were *Microbacterium* (three strains), *Curtobacterium*, *Pseudomonas* (two strains), *Ralstonia*, *Methylobacterium* (two strains) and *Bacillus* (Figure 2).

The taxonomic classification at the species level of the different isolated strains would be improved, and it would be necessary to combine other conserved genes for these differentiations, in addition to the use of the 16S rRNA gene, which is the molecular marker widely used for the taxonomic classification of prokaryotes [36]. Thus, one of the frequently isolated families of endophytes in the laboratory corresponds to the *Microbacteriaceae* family [50], such as our isolates.

On the other hand, some of the different genera isolated correspond to microorganisms with pathogenic capacity. In the case of *Pseudomonas*, depending on the type of host, it can be pathogenic or not. This bacterium can cause disease in some plants while, in others, it can live as a symbiont, developing biocontrol and growth promotion functions. In the case of humans, especially *Pseudomonas aeruginosa*, it is generally known as a pathogen of clinical importance [48].

In addition, the genus *Methylobacterium* was identified. Mostly facultative methylotrophic bacteria with pink pigmentation, this genus plays an essential role in the growth and yield of plants through the synthesis of carotenoids and the production of phytohormones [51].

Another finding of this study was the isolation of the genus *Ralstonia*, which is widely used in bioremediation due to its ability to decompose many toxic substances. Furthermore, this genus develops biocontrol functions like *Microbacterium* and *Bacillus*. The *Bacillus* genus, in particular, also presents antifungal activity for a pathogen that affects cocoa plantations [52].

Some studies isolated and identified endophytic bacteria belonging to the genera *Streptomyces*, *Micronospora*, *Actinomadura*, *Bacillus*, *Paenibacillus*, and *Pseudomonas* [3,6,53].

### 3.3. Antimicrobial Activity

For the bioactive molecules produced, it was necessary to know the growth kinetics of these bacteria, and it was found that the stationary phase began at 16 h, as reported by Almajali et al. [54], for the genus *Curtobacterium* [55]. There is not enough scientific evidence about the stationary phase time of *Microbacterium*. 

The evaluation of the antimicrobial activity of the isolated actinobacteria was visualized by the formation of inhibition halos produced by all strains (CP1B01, CP1A02, CP3A01, and CP3A02) against *Staphylococcus aureus* of the reference and clinical strains. However, for *E. coli* of the reference and clinical strains, *K. pneumoniae*, *P. aeruginosa*, *E. cloacae*, *A. baumanii*, and *S. marcescens*, the same formation was not observed.

The diameters of the inhibition halos formed by *Microbacterium* and *Curtobacterium* isolates against *S. aureus* of the reference and clinical strains are presented in Table 2.

A similar result showed that an actinobacteria, isolated from *Ficus minhasae*, inhibited the growth of *S. aureus* but not *E. coli* [44]. The importance of finding new molecules against *S. aureus* is due to their clinical relevance and their complications in multidrug resistance, which are a frequent health problem, allowing new possibilities for the treatment of methicillin-resistant *S. aureus* (MRSA) strains [56].

There are not enough articles about the use of these antibiotic-producing bacteria against clinical strains. The discovery of this type of biomolecule helps in the development of new treatment alternatives against multi-resistant strains, which can combine synthetic molecules and natural extracts [57].

In brief, actinobacteria are commonly used in the pharmaceutical industry to search for antimicrobial compounds. Due to their non-pathogenic characteristics for humans, they are used for these purposes [58,59].

## 4. Conclusions

In this study, it was possible to isolate 11 endophytic bacterial strains collected from the leaves and stem fragments of cocoa trees. The morphological and biochemical characterization of the isolates was varied. In the case of molecular identification, it was possible to name all strains at the genus level. Of these, four belonged to the actinobacteria group, three to the genus *Microbacterium*, and one to *Curtobacterium*. Other isolates were characterized as *Pseudomanas*, *Bacillus*, *Ralstonia*, and *Methylobacterium*.

Actinobacteria inhibited the growth of *Staphylococcus aureus*, proving to be effective against this bacterium. Therefore, to better understand the study of endophytes in this type of plant, it is necessary to increase the number of cocoa samples. Likewise, with these isolated strains, the extraction and chemical characterization of secondary metabolites with antibacterial potential could be carried out, in addition to being able to develop genomic analyses for functional annotation of the genes responsible for the synthesis of these antimicrobial compounds and, finally, to make comparisons in existing databases.

## Figures and Tables

**Figure 1 microorganisms-13-01686-f001:**
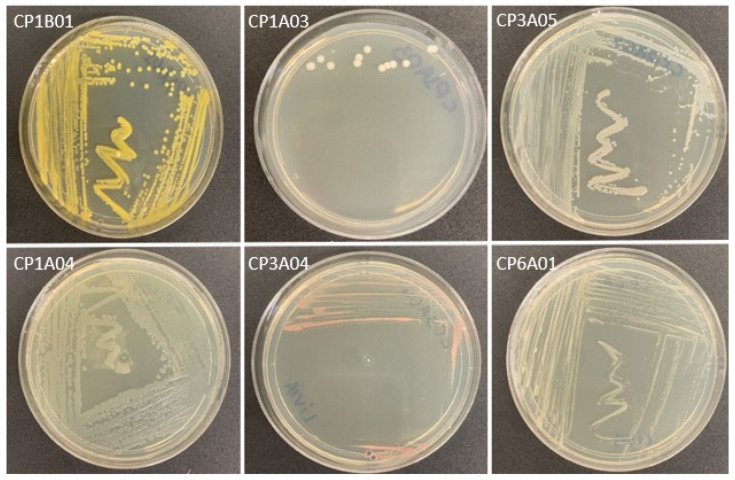
Macroscopic colony characteristics of endophytic bacterial isolates of *Theobroma cacao* on ISP-2 agar.

**Figure 2 microorganisms-13-01686-f002:**
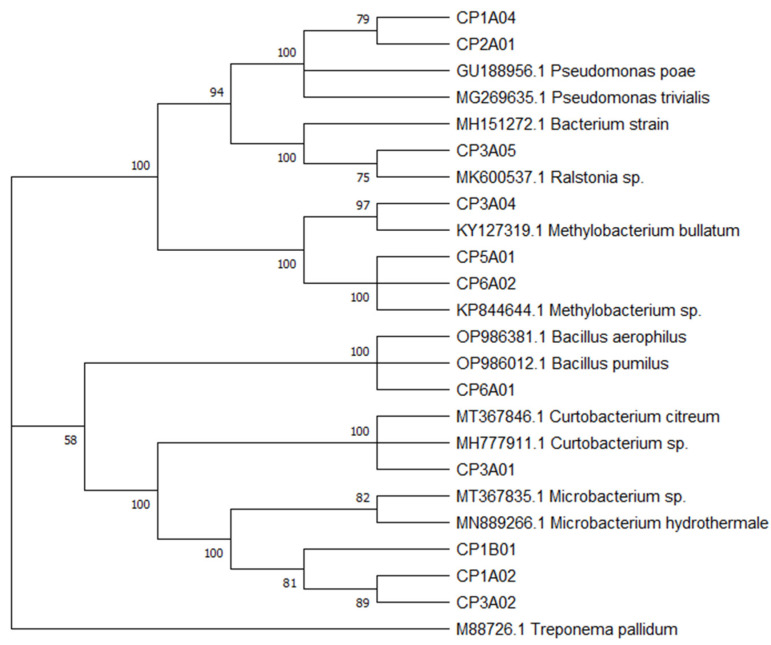
Phylogenetic tree of cultivable endophytic bacteria in *T. cacao*. Phylogeny based on 16S rRNA sequences obtained from isolates and the NCBI database, performed using the Maximum Likelihood method with the Jukes–Cantor model. Associated taxa were grouped in the bootstrap test (1000 replicates), and bootstrap values were greater than 50%. Evolutionary analyses were conducted in MEGA v. 11 software.

**Table 1 microorganisms-13-01686-t001:** Morphological and biochemical characterization of endophytic bacteria strains from *Theobroma cacao*.

Bacterial Code	Morphological Characterization		Biochemical Characterization					
	Macroscopy	Microscopy	Catalase	Citrate	Phenylalanine	Carbohydrates	CO_2_	H_2_S
CP1B01	Circular and convex yellow colonies	Gram-positive rod	+	_	_	Glucose	+	_
CP1A02	Circular and convex yellow colonies	Gram-positive rod	+	_	_	Glucose	+	_
CP1A04	Irregular and flat non-pigmented colonies	Gram-negative rod	_	+	_	Glucose	+	_
CP2A01	Irregular and convex non-pigmented colonies	Gram-negative rod	+	_	_	Glucose	+	_
CP3A01	Circular and convex yellow colonies	Gram-positive rod	+	_	_	Glucose	_	_
CP3A02	Circular and convex yellow colonies	Gram-positive rod	+	_	_	Glucose	_	_
CP3A04	Irregular and umbilicated pink colonies	Gram-negative rod	+	_	_	Glucose	+	_
CP3A05	Circular and flat non-pigmented colonies	Gram-negative rod	+	+	_	Absence	+	_
CP5A01	Irregular and flat pink colonies	Gram-negative rod	+	_	_	Glucose	_	_
CP6A01	Irregular and umbilicated non-pigmented colonies	Gram-positive rod	+	+	_	Glucose/Sucrose/Lactose	+	_
CP6A02	Circular and convex pink colonies	Gram-negative rod	+	_	_	Absence	+	_

**Table 2 microorganisms-13-01686-t002:** Characteristics of actinobacteria strains for evaluation of antimicrobial compound production.

Bacterial Code	Genera	Bacteria Used in the Antagonism Assay			
		*S. aureus* ATCC		*S. aureus* Clinical Strain	
		Inhibition Halos	Standard Deviation	Inhibition Halos	Standard Deviation
CP1B01	*Microbacterium*	26.53 mm	1.66	26.2 mm	2.3
CP1A02	*Microbacterium*	27 mm	4.35	28.53 mm	1.07
CP3A01	*Curtobacterium*	29.73 mm	1.25	30.16 mm	0.53
CP3A02	*Microbacterium*	27.4 mm	3.20	26.96 mm	3.78

## Data Availability

The data presented in the study was deposited in the National Center for Biotechnology Information NCBI database (https://www.ncbi.nlm.nih.gov/), accession ID PQ412810, PP83992-P839001.

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
