# Peer review of "Endophytic Bacteria with Potential Antimicrobial Activity Isolated from Theobroma cacao in Brazilian Amazon"

_microorganisms, 2025, doi:10.3390/microorganisms13071686_

Round 1

Reviewer 1 Report

Comments and Suggestions for Authors

This is an interesting study. There are suggestions that further improvements will be needed to enhance it carefully.

1. Please add quantitative data in the abstact part.

2. what is the implication of your study add 1-2 sentences in the end of abstract part.

3.The introduction part is very short and need more improvement. Strengthen your writing in introduction part; what is the significance of your study; what are your aims or objectives; also add previous latest references. what are the gaps in the previous studies and why are you doing this research?

4.Ensure that each result has been discussed in the discussion part.

5. add some results in the conclusion part.

6. what are the future persceptive of your study add in the conclusion part.

4. 

Author Response

Dear Professor,

Please see the attachment, it contains the responses to your comments.

Reviewer 2 Report

Comments and Suggestions for Authors

Dear authors, I have several comments regarding your article.

1. Please rewrite the introduction. At the moment, the paragraph about secondary metabolites is not related to anything. Also, the phrase about "the isolation of endophytic bacteria associated with native tropical plants is unexplored" is no longer relevant, since you are citing a 2010 article, and there are several articles even about cocoa endophytes.

2. Line 84 - Please write the composition of the saline solution.

3. Why was the antimicrobial activity of other types of bacteria identified in the work not studied?

4. Please add to the discussion a comparison of the species composition of the endophytes identified by you and in other studies.

Author Response

(The authors gave the same response as above.)

Reviewer 3 Report

Comments and Suggestions for Authors

Endophytic bacteria often inhabit leaves, stems and roots of many plants. Numerous species of endophytic bacteria have been found that have an antagonistic effect on pathogenic fungi and are used in biocontrol and biotechnology. The current manuscript presents endophytic bacteria in Theobroma cacao. The title of the current paper requires a slight change. The total number of leaves and stem fragments from which bacteria were isolated was six. This is therefore a small sample. Hence, the comparison between shaded plantation and full sun plantation of Theobroma cacao has major limitations. In the Results and Discussion section, the results should be presented in a more explicit and orderly manner. Currently, the difference between own results and discussion is blurred. Sometimes the discussion precedes own results. The methodology of laboratory and molecular studies is correct. In leaves and stems of T. cacao bacteria representing six genera were found: Microbacterium, Curtobacterium, Pseudomonas, Ralstonia, Methylobacterium and Bacillus. These are interesting results. Also valuable are the results concerning the properties of some bacteria against Staphyllococcus aureus. The discussion is written in an interesting way, it is well supported by the literature. After correction, the manuscript should be published in Microorganisms.

Remarks

Line 2-3 I think the title should be slightly changed, for example: Endophytic bacteria with potential antimicrobial activity isolated from Theobroma cacao

Line 29-32 consider revising this sentence

Line 30 Pseudomonas - it should be in italic

Line 38 Theobrama - it should be Theobroma

Line 74 leaves and stem fragments were collected - numbers should be given

Line 75 The collected material was identified - what does this mean? Collected material was leaves and stem fragments - this material required identification? - this is unclear

Line 81-82 needs to be completed - was it ethyl alcohol?

Line 131 it should be Klebsiella pneumoniae

Line 149 you write : 19 endophytic bacterial colonies were obtained, while in Table 1  11 strains are characterized - this requires explanation

Line 163 Andrade [6]- it should be Andrade et al. [6]

Line 186 It should be Citrullus colocynthis (L.) Schrad.

Line 261 it should be K. pneumoniae

Line 263-269 - an attempt should be made to present these results in a table

Line 267  S. aureus - it should be in italic

Line 296 Methylobacteriu - this requires correction

Line 298  S. aureus - it should be in italic

Author Response

(The authors gave the same response as above.)

Round 2

Reviewer 2 Report

Comments and Suggestions for Authors

Dear colleagues, please check the introduction again. For example, you are referring to an article that is not devoted to the study of stomata in roots (lines 52-53), but which in the review writes about stomata on leaves and refers to another article. Correct the text and refer to the original article. Also, the word "They" (line 53) refers to microorganisms in the text, but should refer to plants.

Lines 68-70 A reference to a work on reducing the genome of endophytes is needed.

Author Response

Dear, Professor
Thank you for your comments and suggestions. We hope the changes meet your expectations. All this helps us to improve the text so that it can be published in the Journal.

Comment 1: You are referring to an article that is not devoted to the study of stomata in roots (lines 52-53), but which in the review writes about stomata on leaves and refers to another article. Correct the text and refer to the original article.

Response 1: The original article has been consulted and the text has been corrected (lines 52-53).

Comment 2: The word "They" (line 53) refers to microorganisms in the text, but should refer to plants.

Response 2: The word was replaced by the appropriate term (lines 53-54).

Comment 3: Lines 68-70 A reference to a work on reducing the genome of endophytes is needed.

Response 3: It was incorporated in the line 70.